# Relative Efficacy and Safety of Anti-Inflammatory Biologic Agents for Osteoarthritis: A Conventional and Network Meta-Analysis

**DOI:** 10.3390/jcm11143958

**Published:** 2022-07-07

**Authors:** Yang Li, Yiying Mai, Peihua Cao, Xin Wen, Tianxiang Fan, Xiaoshuai Wang, Guangfeng Ruan, Su’an Tang, Changhai Ding, Zhaohua Zhu

**Affiliations:** 1Clinical Research Centre, Zhujiang Hospital, Southern Medical University, Guangzhou 510280, China; watsonhanks@foxmail.com (Y.L.); maiyiying123@163.com (Y.M.); cphcc@smu.edu.cn (P.C.); lcyswenxin@163.com (X.W.); fmuftx@163.com (T.F.); drwangxs2019@126.com (X.W.); ruan1989.ok@163.com (G.R.); tangsan@mail2.sysu.edu.cn (S.T.); changhai.ding@utas.edu.au (C.D.); 2Department of Rheumatology and Clinical Immunology, Zhujiang Hospital, Southern Medical University, Guangzhou 510280, China

**Keywords:** osteoarthritis, biological therapy, inflammation, infliximab, meta-analysis

## Abstract

Previous studies have consistently revealed that both local and systemic inflammations are the key to the onset and progression of osteoarthritis (OA). Thus, anti-inflammatory biologic agents could potentially attenuate the progression of OA. We conducted this meta-analysis to examine the efficacy and safety of ant-inflammatory biologic agents among OA patients. Methods: Five databases were searched for randomized controlled trials (RCTs) comparing biologics with placebo or each other in OA patients. Data of pain, physical function, stiffness, and adverse events (AEs) were extracted for a conventional and a Bayesian network meta-analysis. Results: 15 studies with data for 1566 patients were analyzed. In the conventional meta-analysis, etanercept (SMD −0.47; 95% CI −0.89, −0.05) and infliximab (SMD −2.04; CI −2.56, −1.52) were superior to placebo for knee pain. In the network meta-analysis, infliximab was superior to all the other biologic agents in improving pain (vs. hyaluronic acid (SMD −22.95; CI −34.21, −10.43), vs. adalimumab (SMD −21.71; CI −32.65, −11.00), vs. anakinra (SMD −24.63; CI −38.79, −10.05), vs. canakinumab (SMD −32.83; CI −44.45, −20.68), vs. etanercept (SMD −18.40; CI −29.93, −5.73), vs. lutikizumab (SMD −25.11; CI −36.47, −14.78), vs. naproxen (SMD −30.16; CI −41.78, −17.38), vs. tocilizumab (SMD −24.02; CI −35.63, −11.86) and vs. placebo (SMD −25.88; CI −34.87, −16.60)). No significant differences were observed between biologics and placebo regarding physical function, stiffness, and risk of AEs. Conclusions: The findings suggest that infliximab may relieve pain more than other biological agents in OA patients. No significant differences were observed between biologics and placebo regarding physical function, stiffness, and risk of AEs. The results must be interpreted cautiously; therefore, further randomized controlled trials are warranted.

## 1. Introduction

Osteoarthritis (OA) has become a major health challenge around the world due to its rising prevalence and enormous burden caused individually and socially. There are no approved drugs with disease-modifying effects, let alone the number of risk considerations for the available medications that could relieve symptoms [1,2,3]. Thus, developing new drugs to address unmet medical needs is crucial.

Although OA used to be considered as a noninflammatory disease, it is now well recognized that chronic and low-grade inflammation is involved in OA progression. Inflammatory factors and chemokines have been reported to contribute to inflammation in both synovial cells and chondrocytes [4]. Anti-inflammatory biologic agents, including but not limited to TNF-α inhibitors (e.g., adalimumab), interleukin-1 (IL-1) inhibitors (e.g., canakinumab), and IL-6 inhibitors (e.g., tocilizumab), have been used in treating rheumatoid arthritis (RA) and other inflammatory diseases. They can suppress specific components of the immune system and thereby inhibit the activation of inflammatory pathways mediated by inflammatory factors [5,6,7]. For this reason, anti-inflammatory agents may be promising agents to attenuate disease progression of OA. However, randomized controlled trials (RCTs) examining the efficacy and safety of biologics in OA patients have shown inconclusive results. For example, Fleischmann et al. [8] suggested that lutikizumab use significantly relieved pain compared with placebo, while Kloppenburh et al. [9] reported that lutikizumab did not alleviate pain or imaging outcomes in comparison to placebo. Moreover, previous systematic reviews also indicated the inconsistent efficacy of biologic agents [10,11,12]. Therefore, we performed an up-to-date network meta-analysis to compare the efficacy and safety of biologics targeting inflammation among OA patients. By using network meta-analysis, we can estimate the efficacy and safety between all possible pairs of treatments and then rank them in order of the size of effects. 

## 2. Methods

### 2.1. Protocol and Registration

This network meta-analysis was performed according to the checklist of the Preferred Reporting Items for Systematic Reviews and Meta-analysis (PRISMA) extension statement for network meta-analysis (Appendix A) [13], and registered in PROSPERO (CRD42020196343).

### 2.2. Search Strategies and Selection Criteria

A systematic search was conducted using electronic databases of Medline (PubMed), Embase, Web of Science, Cochrane Library Central Register of Controlled Trials (CENTRAL), and www.ClinicalTrials.gov, from inception to 1 February 2022. The language was limited to English. The search procedures and strategies are shown in Appendix A. Eligible studies met the following criteria:RCTs.Patients with clinically or radiographically diagnosed primary OA at any joints.Interventions or exposures included adalimumab, lutikizumab (ABT981), canakinumab, anakinra, etanercept, infliximab, and any other TNF-α, IL-1, IL-6 or IL-17 inhibitors alone or in combination.

Studies met the following criteria were excluded:Retrospective research, review, or meta-analysis.Studies that only published as abstract or without extractable data.Follow-up duration <1 week.Studies that did not report pain, physical function, stiffness, or adverse events (AEs) as outcomes.

All retrieved articles were imported into EndNote X9 software. After excluding repeated ones, two investigators (YL and YM) screened the titles and abstracts independently according to the inclusion criteria. Any disagreements between them over the eligibility of particular studies were resolved through discussion with a third investigator (ZZ).

## 3. Outcomes and Data Extraction

The primary outcomes were mean changes from baseline in pain and physical function score. Secondary outcomes were AEs and mean change in stiffness score. For pain, physical function, and stiffness, the time point was at or nearest to 12 weeks, and for Aes, the time point was at the end of the study. When pain, physical function, and stiffness were measured using different scales in one study, we referred to a previously described hierarchy of relative outcomes and extracted data that was highest on the list (Appendix A) [14,15].

Two investigators (YL and YM) extracted data independently with standardized forms. The data were checked by a third investigator (ZZ). For pain, physical function, and stiffness, the changes from baseline at or nearest to 12 weeks were extracted and calculated as the arithmetic differences between baseline and follow-up. If standard deviations (SD) were not provided, we calculated or imputed them using methods reported in Cochrane Handbook for Systematic Reviews of Interventions [16]. For AEs, the number of patients who experienced any AEs and withdrawal due to AEs was calculated. For graphical information, numerical data was extracted using Engauge Digitizer 12.1 software (Mark Mitchell, Palos Verdes Peninsula, CA, USA). If a study involved multiple treatment groups with different doses or administration of the same drugs, the data were combined into one treatment group.

For crossover studies, if the data were given based on the order in which the participants received the treatments, the data from each period were extracted and analysed separately. Other extracted data included first author, year of publication, study design, details of interventions, sample size, demographic characteristics [age, sex, and body mass index (BMI)], follow-up duration, study joint, and outcome assessment.

## 4. Quality Assessments

Two investigators (YL and YM) assessed the risk of bias of included studies independently using the Cochrane Risk of Bias Tool for RCT and the Newcastle-Ottawa scale (NOS) for prospective cohort study [17,18]. The Cochran Risk of Bias Tool for RCT assessed five aspects: random sequence generation, allocation concealment, blinding method, outcome assessment, and reporting of result. Each aspect was judged to be low, unclear, or high risk of bias. For NOS, selection of the study groups, comparability among different groups, and ascertainment of either the interested exposure or outcome were evaluated. A score less than 4 indicates a high risk of bias; a score between 4 and 6 indicates a moderate risk of bias; and a score equal to or higher than 7 indicates a low risk of bias.

## 5. Statistical Analyses

To estimate the pooled odds ratios (OR) for dichotomous outcomes and standardized mean differences (SMD) for continuous variables, we first performed a conventional meta-analysis with RevMan 5.4 software (Cochrane Collaborating, Copenhagen, Denmark). Heterogeneity in each direct comparison was assessed using the *I*^2^ test (*I*^2^ ≥ 50% was considered heterogeneous and a random-effect model was used, otherwise a fixed-effect model was used). Sensitivity analyses were conducted to test the robustness of the results under the fixed and random models. Subgroup analyses were also conducted if applicable.

A Bayesian network meta-analysis was then performed using ADDIS 1.16.5 software (Drug Information Systems, Groningen, The Netherlands) [19]. Based on the Markov chain Monte Carlo (MCMC) simulation method, the Bayesian network meta-analysis method can integrate all direct and indirect comparisons and estimate the probability of each intervention becoming the best one. The consistency between direct and indirect comparisons was tested by node-splitting analysis and inconsistency standard deviation (ISD). When node-splitting analysis determined a *p* value > 0.05 and 95% CI of ISD included 1, the consistency model was used for pooled analysis; otherwise, the inconsistency model was used [20]. The model convergence was assessed using a potential scale reduction factor (PSRF) of the Brooks-Gelman-Rubin (BGR) diagnostic [21]. PSRF closer to 1 indicated better convergence, and it was acceptable if PSRF < 1.2. Finally, the ranking probability of agents for each outcome was calculated.

Stata 15.1 software (Stata Corp, College Station, TX, USA) was used to draw the network plot and assess the publication bias by examining funnel plot asymmetry and Egger’s test. A roughly symmetrical funnel plot and an Egger’s test *p* value over 0.05 indicates no evidence of publication bias.

## 6. Results

A total of 758 records were retrieved, of which 15 RCTs met the predefined criteria, including 1566 patients (Figure 1) [8,9,22,23,24,25,26,27,28,29,30,31,32,33,34]. No observational study was included. Table 1 showed the baseline demographic characteristics of included studies. Eight studies included patients with knee OA and seven studies included patients with hand OA. The mean age of included patients ranged from 54.3 to 66.0 years. All the patients were categorized into 12 intervention groups according to different treatments they had received: placebo, adalimumab, lutikizumab (ABT981), canakinumab, naproxen, hyaluronic acid (HA), anakinra, etanercept, infliximab, AMG108, tocilizumab, and standard care. Naproxen and HA were the control groups in some of the studies and were therefore included in the analysis. AEs were reported in all the studies and 12 studies reported outcome measures for pain, nine for physical function, and six for stiffness. Baseline characteristics of patients were generally comparable regarding age, sex composition, BMI, OA severity, and disease duration within studies.

Significant pain reductions were found in the following comparisons of conventional meta-analysis (Appendix A): etanercept vs. placebo (SMD −0.47; −0.89 to −0.05), infliximab vs. placebo (SMD −2.04; −2.56 to −1.52), tocilizumab vs. placebo (SMD −0.60; −1.05 to −0.15), and adalimumab vs. HA (SMD −0.62; −1.16 to −0.09). In addition, canakinumab showed a weaker analgesic effect compared with both placebo (SMD −1.69; −1.2 to −2.19) and naproxen (SMD −0.66; −0.24 to −1.08). According to the network meta-analysis (Table 2 and Figure 2), Infliximab was associated with significantly more pain reduction than all the other drugs: infliximab vs. HA (SMD −22.95; −34.21 to −10.43), infliximab vs. adalimumab (SMD −21.71; −32.65 to −11.00), infliximab vs. anakinra (SMD −24.63; −38.79 to −10.05), infliximab vs. canakinumab (SMD −32.83; −44.45 to −20.68), infliximab vs. etanercept (SMD −18.40; −29.97 to −5.73), infliximab vs. lutikizumab (SMD −25.11; −36.47 to −14.78), infliximab vs. naproxen (SMD −30.16; −41.78 to −17.38), infliximab vs. placebo (SMD −25.88; −34.87 to −16.60), infliximab vs. tocilizumab (SMD −24.02; −35.63 to −11.86). Adalimumab (SMD −11.11; −20.16 to −1.26) and etanercept (SMD −14.40; −26.10 to −3.24) were significantly better than canakinumab. And etanercept have a stronger analgesic effect than naproxen (SMD −11.71; −23.06 to −0.52). But adalimumab, naproxen, and etanercept were both not superior to placebo. All other comparisons did not show significant differences. Similar result was found in probability ranking (Appendix A), which indicated that infliximab was the best drug (98% chance) for analgesia, while canakinumab (79% chance) was the worst.

In the conventional meta-analysis for physical function (Appendix A), adalimumab was associated with a greater physical function improvement compared with HA (SMD −0.88; −1.44 to −0.33), and tocilizumab can significantly improve function compared with placebo (SMD −1.48; −2.00 to −0.97). However, canakinumab showed a weaker physical function improvement compared to placebo (SMD −1.55; −1.07 to −2.03) and naproxen (SMD −0.52; −0.10 to −0.94). No significant difference was found in other comparisons. None of the drugs showed significant differences compared with placebo in network meta-analysis (Table 3), while probability ranking provided the hierarchy of physical function-improving effect and indicated that etanercept (28% chance) could be the best option for function improvement (Appendix A).

In terms of stiffness, the conventional meta-analysis demonstrated that canakinumab was associated with a weaker stiffness improvement compared to placebo (SMD −1.61; −1.12 to −2.09) and naproxen (SMD −0.83; −0.40 to −1.25). The remaining interventions were not associated with significant improvement in stiffness (Appendix A). Network meta-analysis demonstrated no significant differences in all comparisons (Appendix A). Based on the probability ranking, lutikizumab (37% chance) was the best option for stiffness, while etanercept (30% chance) was the worst (Appendix A).

All studies reported outcomes of AEs. No significant difference was reported regarding incidence rates of AEs between treatment and control groups (Appendix A). AEs were common in most studies except for that two studies [26,28] did not record any AEs and one study [31] recorded only one patient developed AEs. Reported AEs included fall, headache, infections, sinusitis, vertigo, eczema, rash, or itching, injection site reaction, neutropenia, malignancies, and death. The most frequently reported AEs were infections, injection site reaction, and arthralgia. Yet, serious AEs were rare. Dose-dependent increases in AEs were found in anakinra and lutikizumab [9,23].

Three studies [26,28,31] were excluded from the network meta-analysis of AEs to prevent a widely pooled confidence interval and inaccurate results because their number of AEs in the treatment group and/or the control group were zero. An inconsistency model was used for network meta-analysis because the calculated 95% CI of ISD (ISD 0.43; 0.03 to 0.82) did not include 1. No significant results were found in the conventional and network meta-analysis (Appendix A and Appendix A), suggesting that anti-inflammatory biologics did not increase AEs. Rank probability was not available in the inconsistency model.

The quality assessments for pain, physical function, stiffness and adverse events indicated no serious risk of bias (data not shown). Figure 3 shows the quality assessment for adverse events. 53% of the studies were judged to have a low risk of bias for random sequence generation, 47% for allocation concealment, 93% for incomplete outcome data, 60% for blinding of participants, 93% for selective reporting, and 73% for blinding of outcome assessment. Two studies (13%) were judged to have a high risk of bias for random sequence generation since they did not mention randomization; two (13%) for allocation concealment since their allocation results could be predicted; one (7%) for incomplete data since it only analysed the completers’ data; one (7%) for blinding of participants since it was an open-label design, and one (7%) for selective reporting since it did not fully report the outcomes. All studies had unclear risks of other bias because they could not be judged clearly.

Sensitivity analyses were conducted to test the robustness of the results under the fixed model and random model, and no change was revealed (Appendix A). Except for the AEs comparison, which showed a significant inconsistency, the homogeneity and consistency assumptions of the remaining outcomes comparisons were confirmed (Appendix A). The funnel plot (Appendix A) and the Egger’s test (Appendix A) found no publication bias. Subgroup analyses by OA locations (hand and knee) using conventional meta-analysis did not show statistically significant symptoms relief by comparing biologics with placebo (data not shown).

## 7. Discussion

We estimated the relative efficacy and safety of novel biologics targeting inflammations for the treatment of OA using network meta-analysis. Despite limited sample size, we found that infliximab was the most effective treatment compared with all other biologics regarding pain relief. Moreover, according to conventional meta-analysis, etanercept was associated with greater pain relief, and tocilizumab was associated with improvement in pain and physical function, compared with placebo. All the biologics did not increase AEs and were tolerable for OA patients.

The efficacy and safety of anti-inflammatory biologics have been widely studied in other inflammatory diseases. Multiple clinical trials found that infliximab, an anti-TNF-α biologics, was effective for ankylosing spondylitis (AS) and RA [35,36]. Sbidian et al. reported that biologics targeting IL-17, IL-12/23, and TNF-α were more effective than placebo while retaining a sound safety profile for the treatment of psoriasis [37]. A meta-analysis also confirmed the efficacy of anti-TNF-α biologics for inducing and maintaining mucosal healing in patients with Crohn’s disease and ulcerative colitis [38]. By using pooled analysis of the latest clinical trials based on the MCMC simulation method, we can retain direct effects of treatments in each trial and compare all the treatments across trials with a sound statistical precision at the same time. Our study found that infliximab achieved a greater pain relief than any other biologics or placebo, yet it did not increase AEs. Infliximab is a monoclonal IgG1 antibody against TNF-α. It exerts an anti-inflammatory effect by directly binding to TNF-α and blocking its affinity with the corresponding receptors [39]. Our finding suggests that targeting TNF-α could also be an effective therapeutic strategy for OA.

In contrast, other types of TNF-α inhibitors (e.g., adalimumab and etanercept) were not significantly associated with improved OA symptoms. Adalimumab and etanercept exert anti-inflammatory effects through the same mechanism as infliximab, but with a different antibody-protein composition [40]. One possible reason for the inconsistent efficacy of TNF-α inhibitors is the presence of anti-drug antibodies (ADAs) [41]. Since biologics are proteins, they can trigger the immune response and induce ADAs formation. ADAs can cause non-response to the treatment and increase the risk of AEs in RA, psoriasis and other inflammatory diseases [42,43,44]. Numerous factors such as molecular structure, dose, sex, and co-administration with other anti-inflammatory drugs, may have influenced the immunogenic of biologics. To the best of our knowledge, there is no study examining how ADAs affect biologic therapies for OA. Hence, more research are needed in the future to disentangle this, and it is vital to consider immunogenic when selecting biologics as the therapy.

Another reason may be related to the way of drug administration. It is reported that free drugs injected in the articular joint can be rapidly cleared, resulting in decreased retention time, low peak drug concentration, and limited therapeutic effect [45]. Given that nearly half of the included studies used intra-articular injection, it was not surprising that our meta-analysis and most of the clinical trials demonstrated negative results. Recently, several advanced drug delivery systems have been developed and proved to be effective in prolonging retention time and improving targeting specificity in animal models [46,47]. Combining novel drug delivery systems and investigational biologics could be an optimal strategy for the treatment of OA, albeit rational designed clinical trials are warranted to validate their efficacies.

Cytokines play important roles in OA progression [48]. However, our network meta-analysis indicated the remaining biologics did not result in symptoms relief compared to placebo. This may be due to the heterogeneity of OA phenotypes and the complexity of the interaction of pro-inflammatory signaling pathways [49,50]. Current meta-analyses found that although biologic agents were generally effective for OA pain relief, subgroup of IL-1 inhibitors or TNF-α inhibitors were not superior to placebo [10,11,12]. We demonstrated consistent results on the ineffectiveness of IL-1 inhibitors, but inconsistently we found infliximab could be effective. It may suggest that the efficacy of biologic agents varies according to mechanism of action, and pro-inflammatory cytokines are not the key drivers of OA symptoms. Meanwhile, only one to two RCTs were performed for each of the remaining agents, suggesting it is too early to jump to a definite conclusion. We notice that there are currently numerous RCTs in progress and it can be inferred that more studies on novel biological interventions targeting inflammation of OA will appear in the next few years.

We try our best to summarize three potential criteria to profile patients that could benefit the most from infliximab treatment. First, since women generally have more inflammation compared to men, infliximab could be more effective in female OA patients [51,52]. Second, a trial has shown that anti-TNFα could halt the progression in OA patients with swollen joint [29]. Thus, OA patients with inflammatory phenotypes such as synovitis and/or effusion could be more suitable for infliximab treatment. Third, when anti-TNFα therapy was applied to erosive hand OA patients who already have cartilage damage, limited improvement was observed for the structure [22], suggesting that infliximab may achieve better efficacy in the early stage of OA.

To verify our findings, we used different models for analysis which all provided consistent results. Moreover, the well-fitted network model and the low-level heterogeneity indicated the robustness and accuracy of the results. However, this meta-analysis is also subject to potential limitations. First, the number of pooled studies was relatively small, and some included studies had limited sample sizes. Our main finding of infliximab was based on only two trials with only 26 included patients, and there were few direct comparisons. Second, we combined groups of different doses and administration methods of the same intervention. We also combined data on hand and knee OA patients. Women could response more actively to anti-inflammatory agents since they have higher OA prevalence and more inflammation. Unfortunately, we were unable to perform further subgroup analysis at the gender level due to limited number of studies. Nevertheless, gender compositions were largely similar across the 12 intervention groups, ranging from 62.4% to 83.3%, suggesting the impact of gender position on efficacy was minimum. Third, estimated SD values and image data extracted by software were used for analyses, which may be inaccurate. However, the estimated SD values were calculated by official methods of Cochrane Handbook for Systematic Reviews of Intervention, and image data were extracted using Engauge Digitizer software, both of which were considered reliable [16,53]. Fourth, we only extracted data at or nearest to 12 weeks for analysis. The analyses may not be generally applicable to other time points. Last, six studies had high risk of methodological bias. But we could not assess the impact of high-risk studies through sensitive analysis, because most comparisons have only one study. Thus, our results must be interpreted with caution.

## 8. Conclusions

The findings suggest that infliximab may relieve pain more than other biological agents in OA patients. No significant differences were observed between biologics and placebo regarding physical function, stiffness, and risk of AEs. The results must be interpreted cautiously; therefore, further randomized controlled trials are warranted.

## Figures and Tables

**Figure 1 jcm-11-03958-f001:**
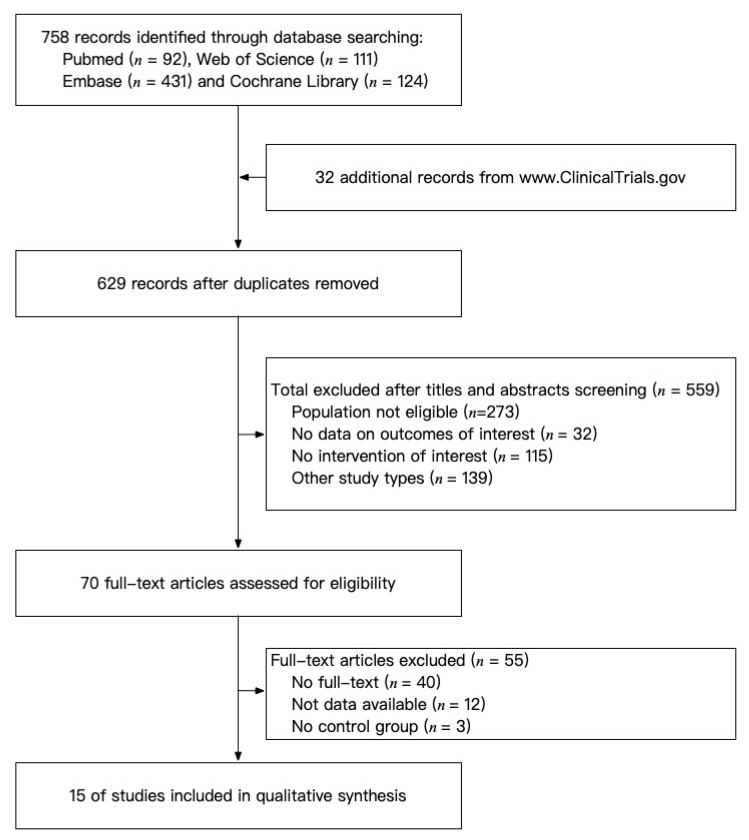
Detailed study selection process.

**Figure 2 jcm-11-03958-f002:**
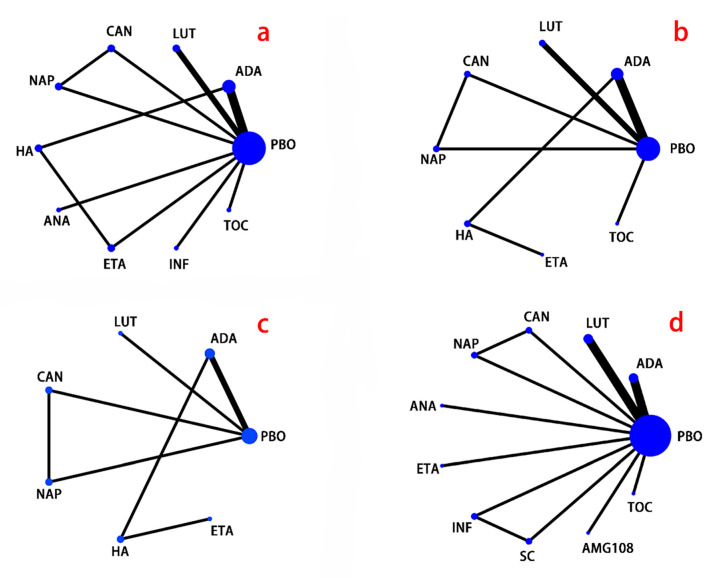
Network plot. (**a**), pain; (**b**), physical function; (**c**), stiffness; (**d**), adverse events; PBO, placebo; ADA, adalimumab; LUT, lutikizumab (ABT981); CAN, canakinumab; NAP, naproxen; HA, hyaluronic acid; ANA, anakinra; ETA, etanercept; INF, infliximab; SC, standard care; TOC, tocilizumab. The size of the circle represents the number of participants, and the thickness of the line represents the number of studies.

**Figure 3 jcm-11-03958-f003:**
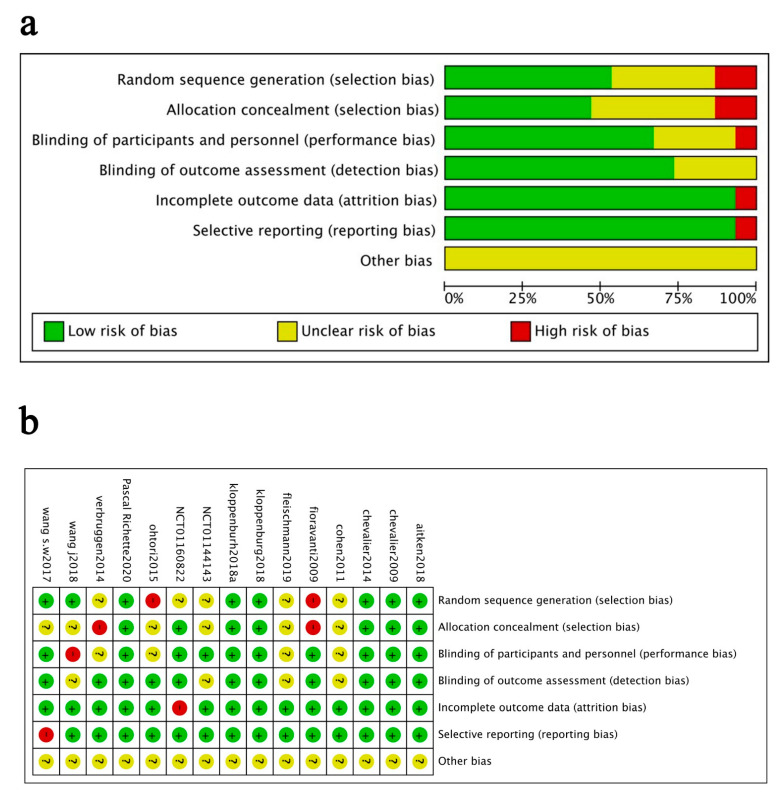
Risk of bias assessment. (**a**) judgements of each bias item presented as percentages across all included studies. (**b**) judgements of each bias item for each included study.

**Table 1 jcm-11-03958-t001:** Characteristics of included studies.

First Author, Publication Year	Study Design	Intervention		Sample Size	Female, *n* (%)	Age (Year)	BMI	Duration of Complaints (Year)	Follow-Up	Joint	Outcome Assessment
D. Aitken, 2018 [22]	Crossover RCT	Placebo		25	18 (72)	61.2 ± 8.4	28.8 ± 4.5	NA	12 weeks	hand	pain, function, stiffness and adverse events.
		Adalimumab (40 mg)		18	15 (83)	63.1 ± 8.4	29.2 ± 3.8	NA			
Chevalier, 2009 [23]	RCT	Placebo		69	44 (64)	62.2 ± 10	NA	6 ± 6.2	12 weeks	knee	pain, function, stiffness and adverse events.
		Anakinra (50 mg)		34	17 (50)	63.3 ± 9.8	NA	8.1 ± 9.8			
		Anakinra (150 mg)		67	46 (69)	62.6 ± 9.4	NA	5.2 ± 5.7			
Chevalier, 2014 [24]	RCT	Placebo		42	35 (83.3)	62.2 ± 7	24.7 ± 3.5	13.5 ± 9.1	26 weeks	hand	function and adverse events.
		Adalimumab (40 mg)		41	36 (87.8)	62.8 ± 6.9	25.2 ± 4.6	13.5 ± 9.8			
Verbruggen, 2014 [29]	RCT	Placebo		30	(83.3)	60.7 ± 6.9	NA	14.4 ± 8.8	52 weeks	hand	pain, stiffness, function and adverse events.
		Adalimumab (40 mg)		30	(86.7)	61.9 ± 6.1	NA	9.6 ± 6.1			
Fleischmann, 2019 [8]	RCT	Placebo		85	52 (61.2)	59.5 ± 8.9	28.6 ± 3.6	7.9 ± 8	52 weeks	knee	pain and adverse events.
		Lutikizumab (25 mg)		89	63 (70.8)	61.6 ± 7.5	28.7 ± 3.8	7.6 ± 9			
		Lutikizumab (100 mg)		85	53 (62.4)	60.2 ± 8.2	29 ± 3.5	7.9 ± 8.7			
		Lutikizumab (200 mg)		88	57 (64.8)	59.1 ± 10.3	28.7 ± 3.5	8.7 ± 8.6			
Kloppenburg, 2018 [9]	RCT	Placebo		67	58 (87)	66 ± 7	28 ± 5	11 ± 8	26 weeks	hand	function; adverse events.
		Lutikizumab (200 mg)		64	53 (83)	66 ± 8	27 ± 5	11 ± 9			
Wang S.X., 2017 [30]	RCT	Part A	Placebo	6	5 (83.3)	60 ± 5.9	28.4 ± 2.3	NA	127 days	knee	adverse events.
			ABT981 (0.3 mg/kg)	7	5 (71.4)	61.3 ± 5.1	27.6 ± 4.4	NA			
			ABT981 (1 mg/kg)	7	5 (71.4)	62.6 ± 3.6	26.4 ± 1.1	NA			
			ABT981 (3 mg/kg)	7	7 (100)	61.4 ± 5	27.3 ± 2.9	NA			
		Part B	Placebo	2	2 (100)	55 ± 1.4	28.7 ± 0.5	NA			
			ABT981 (3 mg/kg)	7	7 (100)	60 ± 6.1	29.3 ± 3	NA			
Kloppenburg, 2018a [27]	RCT	Placebo		45	36 (80)	60.1 ± 8.7	25.5 ± 3.8	10.7 ± 8	1 year	hand	pain and adverse events.
		Etanercept (25–50 mg)		45	37 (82)	59.4 ± 6.5	26.3 ± 3.8	8.8 ± 6			
NCT01144143, 2018 [33]	RCT	Placebo		4	4 (100)	NA	NA	NA	2 months	knee	adverse events.
		Standard care (Methylprednisolone acetate)		4	4 (100)	NA	NA	NA			
		Infliximab		8	5 (62.5)	NA	NA	NA			
NCT01160822, 2012 [32]	RCT	Part A	Placebo	5	2 (40)	57.8 ± 7.8	NA	NA	126 days	knee	pain, stiffness, function and adverse events.
			Canakinumab (150 mg)	6	3 (50)	58.3 ± 12.8	NA	NA			
			Canakinumab (300 mg)	7	4 (57.1)	61 ± 9.6	NA	NA			
			Canakinumab (600 mg)	6	2 (33.3)	64.2 ± 10.7	NA	NA			
		Part B	Placebo	47	31 (66)	60.3 ± 9.7	NA	NA			
			Canakinumab (600 mg)	45	31 (68.9)	61.4 ± 9.0	NA	NA			
			Naproxen (500 mg)	53	34 (64.2)	62.2 ± 8.1	NA	NA			
Cohen, 2011 [25]	RCT	Part A	placebo	16	10 (63)	60.8	30.4	9.6	140 days	knee	Part A: adverse events; Part B: pain, function, stiffness and adverse events.
			AMG108 (100 mg)	12	11 (92)	61.1	30.8	6.9			
			AMG108 (300 mg)	12	7 (58)	62.8	31.9	10.2			
			AMG108 (300 mg)	12	5 (42)	59.6	29.8	6.6			
			AMG108 (75 mg)	12	9 (75)	62.3	30.9	10			
		Part B	Placebo	80	54 (68)	60.1	31.9	6.1	12 weeks		
			AMG108 (300 mg)	80	54 (68)	61.3	32	6.1			
Wang J., 2018 [31]	Open label RCT	HA (25 mg)		28	21 (75)	56.9 ± 9.1	24.7 ± 3.3	NA	4 weeks	knee	pain, function, stiffness and adverse events.
		Adalimumab (10 mg)		28	19 (68)	54.3 ± 8.7	25.3 ± 3.2	NA			
Ohtori, 2015 [28]	RCT	HA (25 mg)		20	13 (65)	64.3 ± 5.6	NA	NA	4 weeks	knee	pain, function, stiffness and adverse events.
		Etanercept (10 mg)		19	13 (68)	63.3 ± 7.2	NA	NA			
Fioravanti, 2009 [26]	RCT	Placebo		10 *	10 (100)	60.7 ± 6.2	NA	7.5 ± 3.5	1 year	hand	pain and adverse events.
		Infliximab (0.2 mg)					NA				
Richette, 2020 [34]	RCT	Placebo		41	34 (82.9)	64.7 ± 8.6	25.7 ± 4.9	10.7 ± 9.8	12 weeks	hand	pain, function and adverse events.
		Tocilizumab (8 mg/kg)		42	34 (81)	64.1 ± 8.9	23.1 ± 3.9	9.1 ± 6.3			

* 10 participants were enrolled in the study. For each patient, the most affected hand was identified and treated with infliximab, while the contralateral hand was treated with placebo. Treatment consisted in injection of infliximab or placebo in each affected proximal interphalangeal and distal interphalangeal joint. The total number of joints treated with infliximab and placebo was 56 and 34, respectively. The number of treated joints was used for analysis. RCT, randomized controlled clinical trial. BMI, body mass index. HA, hyaluronic acid. NA, not avaliable. ABT981, the alias of lutikizumab.

**Table 2 jcm-11-03958-t002:** Network meta-analysis of pain for different interventions.

**HA**	−1.29 (−7.19, 6.82)	1.53 (−11.33, 15.78)	9.86 (−0.86, 21.70)	−4.34 (−12.06, 3.83)	−22.95 (−34.21, −10.43)	2.10 (−6.40, 13.15)	7.07 (−3.31, 18.93)	2.82 (−4.28, 11.63)	0.94 (−9.31, 12.69)
1.29 (−6.82, 7.19)	**adalimumab**	2.61 (−10.00, 15.01)	11.11 (1.26, 20.16)	−3.20 (−12.21, 4.56)	−21.71 (−32.65, −11.00)	3.36 (−4.59, 11.75)	8.32 (−2.04, 17.30)	4.05 (−1.85, 9.67)	2.13 (−7.16, 11.53)
−1.53 (−15.78, 11.33)	−2.61 (−15.01, 10.00)	**anakinra**	8.42 (−5.25, 21.59)	−6.23 (−20.18, 7.22)	−24.63 (−38.79, −10.05)	0.57 (−12.07, 13.76)	5.61 (−7.82, 18.79)	1.37 (−10.19, 12.35)	−0.55 (−13.71, 12.67)
−9.86 (−21.70, 0.86)	−11.11 (−20.16, −1.26)	−8.42 (−21.59, 5.25)	**canakinumab**	−14.40 (−26.10, −3.24)	−32.83 (−44.45, −20.68)	−7.88 (−16.56, 2.73)	−2.76 (−10.55, 4.47)	−7.04 (−14.91, 0.88)	−8.92 (−19.54, 2.60)
4.34 (−3.83, 12.06)	3.20 (−4.56, 12.21)	6.23 (−7.22, 20.18)	14.40 (3.24, 26.10)	**etanercept**	−18.40 (−29.97, −5.73)	6.78 (−2.66, 17.44)	11.71 (0.52, 23.06)	7.49 (−0.57, 15.93)	5.59 (−5.45, 17.28)
22.95 (10.43, 34.21)	21.71 (11.00, 32.65)	24.63 (10.05, 38.79)	32.83 (20.68, 44.45)	18.40 (5.73, 29.97)	**infliximab**	25.11 (14.78, 36.47)	30.16 (17.38, 41.78)	25.88 (16.60, 34.87)	24.02 (11.86, 35.63)
−2.10 (−13.15, 6.40)	−3.36 (−11.75, 4.59)	−0.57 (−13.76, 12.07)	7.88 (−2.73, 16.56)	−6.78 (−17.44, 2.66)	−25.11 (−36.47, −14.78)	**lutikizumab**	5.13 (−6.00, 13.83)	0.79 (−5.86, 6.12)	−1.05 (−11.31, 8.10)
−7.07 (−18.93, 3.31)	−8.32 (−17.30, 2.04)	−5.61 (−18.79, 7.82)	2.76 (−4.47, 10.55)	−11.71 (−23.06, −0.52)	−30.16 (−41.78, −17.38)	−5.13 (−13.83, 6.00)	**naproxen**	−4.34 (−11.64, 3.87)	−6.17 (−16.78, 5.46)
−2.82 (−11.63, 4.28)	−4.05 (−9.67, 1.85)	−1.37 (−12.35, 10.19)	7.04 (−0.88, 14.91)	−7.49 (−15.93, 0.57)	−25.88 (−34.87, −16.60)	−0.79 (−6.12, 5.86)	4.34 (−3.87, 11.64)	**placebo**	−1.86 (−9.55, 5.93)
−0.94 (−12.69, 9.31)	−2.13 (−11.53, 7.16)	0.55 (−12.67, 13.71)	8.92 (−2.60, 19.54)	−5.59 (−17.28, 5.45)	−24.02 (−35.63, −11.86)	1.05 (−8.10, 11.31)	6.17 (−5.46, 16.78)	1.86 (−5.93, 9.55)	**tocilizumab**

HA, hyaluronic acid.

**Table 3 jcm-11-03958-t003:** Network meta-analysis of physical function for different interventions.

**HA**	−11.20 (−27.02, 5.06)	−7.42 (−31.21, 18.42)	−5.56 (−36.82, 26.75)	−12.09 (−33.96, 12.03)	−8.30 (−32.06, 17.23)	−10.49 (−28.85, 9.75)	−11.42 (−34.67, 14.45)
11.20 (−5.06, 27.02)	**adalimumab**	3.79 (−14.76, 23.57)	5.74 (−29.24, 41.42)	−0.71 (−16.17, 16.80)	2.88 (−15.57, 22.17)	0.84 (−9.75, 12.78)	−0.18 (−18.07, 19.53)
7.42 (−18.42, 31.21)	−3.79 (−23.57, 14.76)	**canakinumab**	1.97 (−36.87, 40.63)	−4.61 (−22.84, 14.81)	−0.94 (−15.52, 14.22)	−2.98 (−18.17, 12.22)	−3.96 (−25.31, 17.87)
5.56 (−26.75, 36.82)	−5.74 (−41.42, 29.24)	−1.97 (−40.63, 36.87)	**etanercept**	−6.65 (−44.41, 30.86)	−2.80 (−41.62, 35.74)	−4.73 (−41.97, 31.34)	−5.69 (−45.02, 32.45)
12.09 (−12.03, 33.96)	0.71 (−16.80, 16.17)	4.61 (−14.81, 22.84)	6.65 (−30.86, 44.41)	**lutikizumab**	3.72 (−15.12, 21.86)	1.57 (−10.56, 12.82)	0.68 (−19.27, 19.82)
8.30 (−17.23, 32.06)	−2.88 (−22.17, 15.57)	0.94 (−14.22, 15.52)	2.80 (−35.74, 41.62)	−3.72 (−21.86, 15.12)	**naproxen**	−2.12 (−16.69, 12.69)	−3.03 (−24.39, 18.37)
10.49 (−9.75, 28.85)	−0.84 (−12.78, 9.75)	2.98 (−12.22, 18.17)	4.73 (−31.34, 41.97)	−1.57 (−12.82, 10.56)	2.12 (−12.69, 16.69)	**placebo**	−0.91 (−15.59, 14.26)
11.42 (−14.45, 34.67)	0.18 (−19.53, 18.07)	3.96 (−17.87, 25.31)	5.69 (−32.45, 45.02)	−0.68 (−19.82, 19.27)	3.03 (−18.37, 24.39)	0.91 (−14.26, 15.59)	**tocilizumab**

HA, hyaluronic acid.

## Data Availability

No new data were created in this study. Data sharing is not applicable to this article.

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
