# Peer review of "Relative Efficacy and Safety of Anti-Inflammatory Biologic Agents for Osteoarthritis: A Conventional and Network Meta-Analysis"

_jcm, 2022, doi:10.3390/jcm11143958_

Round 1

Reviewer 1 Report

Overall, I quite liked the review. 

I believe that a better characterization of the patients is pertinent.

I think it is important in the discussion to better explain which criteria (or set) were used for infliximab to be considered the best treatment for OA?

Table 1 needs improvement. there is too much information and reading it becomes tiring and difficult.

Reviewer 2 Report

In this paper, the authors presented the results of a meta-analysis performed to examine the efficacy and safety of anti-inflammatory biologic agents among OA patients. This is a very interesting and up-to-date topic, and the manuscript is clear and well written, presenting detailed described methods and results. I would just advise minor improvements.

Methods: some of the verbs are in the future, consider changing to the past tense.

Results: concerning the sex effects? Was the percentage of women similar in all groups?

Consider discussing this subject.

Figure 2: for the sake of clarity, consider, instead of using numbers, to use the abbreviation of drugs’ name.

Discussion: Although the discussion is well pondered based on the study limitations, and due to these limitations consider rewriting the sentence “Our study found clear evidence that infliximab achieved a greater pain relief than any other biologics or placebo, yet it did not increase AEs”, in particular, consider to remove the expression “clear evidence”.
